# Combined Transarterial Chemoembolization and Radiofrequency Ablation for Hepatocellular Carcinoma Infeasible for Ultrasound-Guided Percutaneous Radiofrequency Ablation: A Comparative Study with General Ultrasound-Guided Radiofrequency Ablation Outcomes

**DOI:** 10.3390/cancers15215193

**Published:** 2023-10-28

**Authors:** Dong Ik Cha, Min Woo Lee, Dongho Hyun, Soo Hyun Ahn, Woo Kyoung Jeong, Hyunchul Rhim

**Affiliations:** 1Department of Radiology and Center for Imaging Science, Samsung Medical Center, School of Medicine, Sungkyunkwan University, 81 Irwon-ro, Gangnam-gu, Seoul 06351, Republic of Korea; dongik.cha@samsung.com (D.I.C.); jeongwk@gmail.com (W.K.J.); hc.rhim@samsung.com (H.R.); 2Department of Health Sciences and Technology, SAIHST, Sungkyunkwan University, 81 Irwon-ro, Gangnam-gu, Seoul 06351, Republic of Korea; 3Department of Mathematics, Ajou University, 206 World Cup-ro, Yeongtong-gu, Suwon 16499, Republic of Korea; pupuppy63@naver.com

**Keywords:** carcinoma, hepatocellular, fluoroscopy, radiofrequency ablation, chemoembolization

## Abstract

**Simple Summary:**

Ultrasound (US) is commonly used as a guiding modality for radiofrequency ablation (RFA) in the treatment of small hepatocellular carcinomas (HCC). However, there are cases where small HCCs are infeasible for RFA under US guidance. In such situations, fluoroscopy-guided transarterial chemoembolization (TACE) combined with RFA (TACE + RFA) may be a potential alternative option. Nevertheless, the long-term effectiveness of TACE + RFA for small HCCs (≤3 cm) infeasible for US-guided RFA has not been thoroughly investigated. This study aimed to evaluate whether or not fluoroscopy-guided TACE + RFA provides comparable outcomes to those of general US-guided RFA. Propensity score (PS) matching analysis was performed, with the feasibility of US-guided RFA excluded from the matching process. The results showed that there were no significant differences in local tumor progression, intrahepatic distant recurrence, and recurrence-free survival between the two groups. Therefore, fluoroscopy-guided TACE + RFA appears to be an effective treatment option when US-guided RFA is not feasible.

**Abstract:**

Objectives: This study aimed to evaluate the therapeutic outcomes of transarterial chemoembolization combined with radiofrequency ablation (TACE + RFA) for hepatocellular carcinomas (HCC) measuring ≤3 cm infeasible for ultrasound (US)-guided percutaneous RFA. Methods: Twenty-four patients who underwent fluoroscopy-guided TACE + RFA for single HCC between January 2012 and December 2016 were screened. To evaluate the TACE + RFA outcomes compared with those of US-guided RFA, 371 patients who underwent US-guided RFA during the same period were screened. We compared local tumor progression (LTP) and intrahepatic distant recurrence (IDR) between the two groups before and after propensity score (PS) matching, and performed univariable and multivariable Cox proportional hazard regression analyses for all patients. Results: PS matching yielded 21 and 42 patients in the TACE + RFA and US-guided RFA groups, respectively. Cumulative LTP rates after PS matching were not significantly different between the two groups at 1 (0.0% vs. 7.4%, *p* = 0.072), 2 (10.5% vs. 7.4%, *p* = 0.701), and 5 years (16.9% vs. 10.5%, *p* = 0.531). IDR rates did not differ significantly between the two groups at 1 (20.6% vs. 10%, *p* = 0.307), 2 (25.9% vs. 25.9%, *p* = 0.999), or 5 years (49.9% vs. 53%, *p* = 0.838). Multivariable analysis showed that treatment type was not a significant factor for LTP or IDR. Conclusion: The outcomes of TACE + RFA for HCC were similar to those of general US-guided RFA. Fluoroscopy-guided TACE + RFA may be an effective treatment when US-guided RFA is not feasible.

## 1. Introduction

Percutaneous ultrasound (US)-guided radiofrequency ablation (RFA) has been widely used to treat small hepatocellular carcinomas (HCCs) (≤3 cm) due to its effectiveness and safety [1,2,3]. However, some HCCs with poor sonic windows or those that are invisible upon US may not be suitable for US-guided RFA. One study reported that US-guided RFA was not feasible in nearly half of the patients that were referred to evaluate the feasibility of US-guided percutaneous RFA [4]. Similarly, small HCCs may not be visible on non-enhanced computed tomography (CT), precluding them from undergoing CT-guided ablation [5]. In these cases, fluoroscopy-guided transarterial chemoembolization (TACE) can be performed for tumor control and to improve patient survival [6], but TACE by itself is not considered a curative treatment, but rather a palliative measure. Instead, fluoroscopy-guided TACE combined with RFA (TACE + RFA) may be a potential alternative treatment option. This is because electrode insertion can be performed under fluoroscopic guidance by targeting the radiopaque iodized oil retained within the tumor from TACE [7]. In addition, the decreased blood flow to tumors from TACE may reduce the heat sink effect, subsequently enabling a larger ablation zone to be created via RFA. In addition, the cytotoxicity of chemotherapeutic agents on cancer cells is enhanced by thermal stress [8] and may enhance treatment effectiveness.

Due to these reasons, previous studies were able to demonstrate the effectiveness of TACE + RFA for intermediate-sized HCCs (3–5 cm) [9,10,11] where the results of RFA alone could potentially have been restricted. However, the long-term effectiveness of TACE + RFA for small HCCs (≤3 cm) not amenable to US-guided RFA has not been thoroughly investigated. A previous study showed that TACE + RFA was superior to TACE monotherapy for HCC cases where US-guided RFA was not feasible [7]. However, whether or not TACE + RFA provides outcomes comparable with those of general US-guided RFA remains unknown. HCCs that cannot undergo US-guided ablation would likewise not be able to undergo US monitoring during TACE + RFA. Consequently, for these particular tumors, the monitoring of the ablation zone during TACE + RFA would have to be dependent on fluoroscopy, which would be very limited. Therefore, it is uncertain whether or not TACE + RFA can achieve treatment results similar to those seen with general US-guided RFA. If TACE + RFA for HCCs infeasible for US-guided RFA proves to be as effective, it may enable the successful ablation of small HCCs that would otherwise be ineligible for RFA, thus broadening the eligibility criteria for patients of ablation therapy.

In this study, we aimed to evaluate the efficacy of fluoroscopy-guided TACE + RFA for the treatment of small HCCs that are infeasible for US-guided RFA. We conducted propensity score (PS) matching analysis to compare the outcomes of TACE + RFA with the general outcomes of US-guided RFA.

## 2. Materials and Methods

### 2.1. Patients

The Institutional Review Board of the Samsung Medical Center approved this retrospective study and waived the requirement for informed consent (IRB No. SMC 2023-07-042). We searched the HCC registry of our medical center to identify patients who underwent TACE + RFA under fluoroscopic guidance between January 2012 and December 2016. This study included patients with the following characteristics: (a) additional factors increasing HCC risk, such as chronic hepatitis B and liver cirrhosis; (b) a single nodular tumor (≤3 cm in size) clinically identified as HCC at the time of treatment; (c) a HCC lesion ineligible for percutaneous US-guided RFA due to tumor invisibility upon US and contrast-enhanced US (CEUS), a poor sonic window, or a poor electrode path; (d) the absence of prior treatment for HCC; (e) the absence of macrovascular invasion or extrahepatic metastasis (EM) as evidenced via pretreatment CT or MRI; (f) Child–Pugh class A or B; (g) no prior or concurrent history of other malignancies; and (h) gadoxetic acid-enhanced liver magnetic resonance imaging (MRI) with diffusion-weighted imaging (DWI) and hepatobiliary phase (HBP) images performed within two months prior to treatment. To assess the comparative efficacy of TACE + RFA against the general outcomes of percutaneous US-guided RFA, we searched our HCC registry for patients who underwent US-guided RFA using identical inclusion criteria, except for criterion (c).

### 2.2. Clinical and Laboratory Factors

Evaluated clinical and laboratory factors included patient age, sex, cause of liver disease, serum albumin level, total bilirubin level, albumin–bilirubin (ALBI) grade, Child–Pugh classification, serum alpha-fetoprotein (AFP) level, serum protein induced by vitamin K absence-II (PIVKA-II) level, serum aspartate aminotransferase (AST) level, platelet count, and AST to platelet ratio index (APRI). The Model for Tumor Recurrence After Living Donor Liver Transplantation (MoRAL) score, which is a scoring system based on serum tumor markers, serum AFP, and PIVKA-II levels, was utilized to calculate the risk of recurrence. Lesions were classified as being at high risk of recurrence if their MoRAL score exceeded the cutoff value of 68 [12].

### 2.3. Image Analysis

Two abdominal radiologists (M.W.L. and D.I.C., with 17 and 7 years of expertise in interpreting liver MRI, respectively) reviewed the pretreatment images of hepatic tumors independently. If any discrepancies arose, they were resolved by a third reviewer (W.K.J., with 17 years of experience in liver MRI interpretation). Features according to the Liver Imaging Reporting and Data System (LI-RADS), such as non-rim arterial phase hyperenhancement (APHE), non-rim washout appearance, enhancing capsule, and LI-RADS category M (LR-M) features were assessed. In addition, tumor size, perivascular location [13], subcapsular location [14], peritumoral parenchymal enhancement in the arterial phase, tumor contour, peritumoral hypointensity upon HBP [15], and lesion signal intensity (SI) upon HBP [16] were evaluated. We utilized a developed scoring system designed to categorize HCCs with a high risk of microvascular invasion (MVI) using AFP, PIVKA-II, peritumoral parenchymal enhancement, and peritumoral hypointensity upon HBP [17]. Details of the models used in this study are provided in the Appendix A.

An experienced radiologist (M.W.L., who possesses 17 years of experience in liver MRI interpretation and tumor ablation) determined tumor size and location. The measurement of tumor size was conducted using the best visible sequence. Regarding tumor location, it was categorized based on its proximity to the liver capsule (subcapsular or non-subcapsular) and the presence of intrahepatic vessels with diameters of 3 mm or larger (portal and hepatic veins). Specifically, a tumor was classified as subcapsular if it was situated within 0.1 cm from the liver capsule.

### 2.4. TACE + RFA and RFA Procedure and Follow-Up Protocol after Treatment

TACE + RFA were performed under fluoroscopic guidance in a single day. Concurrently, US was performed to determine the safe entry site for radiofrequency electrodes and monitor the ablation process. TACE was performed first, immediately followed by RFA, on an inpatient basis by one of three radiologists with more than four years of experience in these procedures. Selective TACE was carried out using a microcatheter by transarterially infusing a mixture of 2–5 mL of iodized oil (Lipiodol; Laboratoire Andre Guerbet, Aulnay-sous-Bois, France) and 10–20 mg of doxorubicin hydrochloride (Adriamycin; Dong-A Pharm, Seoul, Republic of Korea). The iodized oil and doxorubicin was emulsified via vigorous pumping (10–20 times) between two syringes connected with a three-way stopcock just before infusion. The specific amount of iodized oil and doxorubicin administered depended on tumor size and vascularity. Following the transarterial infusion of this mixture, embolization of the tumor’s feeding artery was executed using gelatin sponge pledgets (Cutanplast; Mascia Brunelli, Milan, Italy), which were manually cut into ~1 mm^3^ pieces. Embolization continued until blood flow within the tumor’s feeding artery ceased.

RFA monotherapy was performed under fusion imaging guidance using CT or MR images (volume navigation, LOGIQ E9; GE Healthcare, Chicago, IL, USA) by one of five radiologists with more than three years of experience in locoregional treatments for hepatic tumors. Single or clustered separable electrodes were used with an internally cooled tip (Proteus or Octopus Electrode; STARmed, Goyang, Republic of Korea) or an internally cooled wet tip (Jet-tip; RF Medical, Seoul, Republic of Korea). Multiple overlapping ablations or centripetal RFA using multiple electrodes was performed as required. CEUS using Sonazoid (GE Healthcare, Oslo, Norway) was used to enhance lesion conspicuity if needed. If required, artificial ascites or pleural effusion was introduced to enhance the sonographic window and avoid collateral thermal damage. Multiphase liver CT was performed immediately after RFA to evaluate technical success and procedure-related complications.

Multiphase liver CT and laboratory tests, including those for tumor markers, were performed one month after discharge, every three months for the first two years, and every 4–6 months thereafter. 

### 2.5. Outcome Assessment

The primary outcomes assessed in this study were local tumor progression (LTP) around the ablation zone, intrahepatic distant recurrence (IDR), and recurrence-free survival (RFS). LTP was defined as the detection of disease foci in tumors that were previously deemed completely ablated during the follow-up period [18]. IDR was defined as the development of tumors away from the ablation zone. RFS was defined as the time elapsed between RFA and recurrence or death, wherein recurrence comprised LTP, IDR, and extrahepatic metastasis (EM). EM referred to all tumor lesions diagnosed outside the liver.

### 2.6. Statistical Analysis

The baseline characteristics of the two groups were compared using two-sample t-tests or Mann–Whitney U tests for continuous data depending on normality, and chi-squared tests or Fisher’s exact tests for categorical variables. The effects of selection bias and confounding factors were reduced by calculating PSs using logistic regression, and patients in the TACE + RFA and RFA groups were matched in a 1:2 ratio [19]. The standardized mean difference was computed to check the balance of variables used for matching. Variables demonstrating a difference of *p* < 0.15 between the two groups in the descriptive statistics were selected as covariates for PS matching. Cumulative incidence rate curves for LTP or IDR and Kaplan–Meier curves for RFS with log-rank tests were generated. Additionally, univariable Cox proportional hazard regression analysis was performed, and variables with *p* < 0.15 in the results were included in multivariable analysis using the stepwise selection method to evaluate the association between treatment methods and each outcome. The variables selected for the final model were checked for multicollinearity using a variance inflation factor of less than 10.

Weighted kappa statistics with 95% confidence intervals (CIs) were used to evaluate inter-reader agreement on HCC imaging findings based on an independent image review. Statistical analyses were performed using R version 3.5.0 (The R Foundation for Statistical Computing, Vienna, Austria).

## 3. Results

### 3.1. Patients

During the study period, 24 patients underwent TACE + RFA, and 371 underwent RFA alone. Descriptive data are shown in Table 1. There were 22 men (22/24, 91.7%) in the TACE + RFA group and 272 men (272/371, 73.3%) in the RFA group. The mean age was 57.5 (39–77) years in the TACE + RFA group and 58.0 (31–85) years in the RFA group. The median follow-up period was 73.3 (3.2–113.0) months in the TACE + RFA group and 74.9 (18.1–101.9) months in the RFA group. 

Total bilirubin (*p* = 0.123), prothrombin time (international normalized ratio (INR)) (*p* = 0.092), serum platelet count (*p* = 0.102), AFP levels [log(AFP)] (*p <* 0.001), sex (*p* = 0.079), cause of liver disease (*p* = 0.023), periportal vein tumor location (*p* = 0.013), washout appearance (*p* = 0.112), enhancing capsule (*p* = 0.070), LI-RADS score (*p* = 0.006), and non-smooth tumor margin (*p* = 0.064) varied between the TACE + RFA and RFA groups with *p* < 0.15, and these variables were selected for PS matching. PS matching yielded 21 patients in the TACE + RFA group and 42 patients in the US-guided RFA group (Figure 1), with the variables well balanced between the two groups (Table 2, Appendix A, Figure 2).

### 3.2. Comparison of Therapeutic Outcomes before PS Matching

LTP. LTP occurred in 3 (3/24, 12.5%) patients in the TACE + RFA group and 44 (44/371, 11.9%) patients in the RFA group during follow-up (*p* = 0.965, Figure 3A). Cumulative incidence rates were lower in the TACE + RFA group than in the RFA group at 1 year (0.0% vs. 5.4%, *p <* 0.001) and did not significantly differ at 2 (9.1% vs. 8.6%, *p* = 0.939) or 5 years (15.2% vs. 11.5%, *p* = 0.667). 

IDR. IDR occurred in 11 (11/24, 45.8%) patients in the TACE + RFA group and 167 (167/371, 45.0%) patients in the RFA group during follow-up (*p* = 0.872, Figure 3B). Cumulative incidence rates did not significantly differ between the two groups at 1 (17.8% vs. 10.8%, *p* = 0.392), 2 (22.4% vs. 25.8%, *p* = 0.709), or 5 years (47.8% vs. 45.3%, *p* = 0.828).

RFS. Recurrence or death occurred in 14 patients (14/24, 58.3%) in the TACE + RFA group and 204 patients (204/371, 55.0%) in the RFA group (*p* = 0.753, Figure 3C). RFS did not vary significantly between the two groups at 1 (78.3% vs. 84.6%, *p* = 0.490), 2 (64.9% vs. 66.2%, *p* = 0.902), or 5 years (38.8% vs. 45.6%, *p* = 0.545).

### 3.3. Comparison of Therapeutic Outcomes after PS Matching

LTP. When using matched data, LTP occurred in three (3/21, 14.3%) patients in the TACE + RFA group and five (5/42, 11.9%) patients in the RFA group during follow-up (*p* = 0.895, Figure 4A). Cumulative LTP rates did not significantly differ between the two groups at 1 (0.0% vs. 7.4%, *p* = 0.072), 2 (10.5% vs. 7.4%, *p* = 0.701), or 5 years (16.9% vs. 10.5%, *p* = 0.531). 

IDR. IDR occurred in 10 (10/21, 47.6%) patients in the TACE + RFA group and 22 (22/42, 52.4%) patients in the RFA group during follow-up (*p* = 0.581, Figure 4B). Cumulative IDR rates did not significantly differ between the two groups at 1 (20.6% vs. 10%, *p* = 0.307), 2 (25.9% vs. 25.9%, *p* = 0.999), or 5 years (49.9% vs. 53%, *p* = 0.838). 

RFS. Recurrence or death occurred in 13 (13/21, 61.9%) patients in the TACE + RFA group and 28 (28/42, 66.7%) patients in the RFA group (*p* = 0.829, Figure 4C). RFS was not significantly different between the two groups at 1 year (80.3% vs. 78.6%, *p* = 0.644), 2 years (64.8% vs. 63.1%, *p* = 0.699), or 5 years (35.6% vs. 35.4%, *p* = 0.974).

### 3.4. Multivariable Analysis Using All Patients for Each Outcome

Univariable and multivariable Cox proportional hazard analyses showed that the treatment type (TACE + RFA vs. RFA) was not a significant risk factor for LTP, IDR, or RFS (Table 3, Figure 5). Subcapsular location was a significant risk factor (hazard ratio (HR) = 1.898; 95% CI = 1.071–1.898; *p* = 0.028). For IDR, patient age (HR: 1.023; 95% CI: 1.006–1.023, *p* = 0.009), serum albumin level (HR: 0.477; 95% CI: 0.338–0.477, *p <* 0.001), MoRAL score > 68 (HR: 1.484; 95% CI: 1.051–1.484, *p* = 0.025), and the presence of an enhancing capsule on MRI (HR: 1.536; 95% CI: 1.115–1.536, *p* = 0.009) were significant risk factors. For RFS, age (HR: 1.025; 95% CI: 1.009–1.025, *p* = 0.002), serum albumin level (HR: 0.501; 95% CI: 0.370–0.501, *p <* 0.001), platelet count (HR: 0.995; 95% CI: 0.991–0.995, *p* = 0.030), MoRAL score > 68 (HR: 1.475; 95% CI: 1.099–1.475, *p* = 0.010), and tumor size (HR: 1.653; 95% CI: 1.190–1.653, *p* = 0.003) were significant factors.

### 3.5. Inter-Reader Agreement

All imaging variables exhibited substantial or excellent agreement. Detailed results can be found in the Appendix A. 

## 4. Discussion

Our results show that TACE + RFA under fluoroscopic guidance is an effective treatment for small (≤3 cm) HCCs that are infeasible for US-guided RFA. Since HCCs in the TACE + RFA group were considered infeasible for US-guided RFA, we could not directly compare the outcomes of TACE + RFA with those of the former procedure. Instead, we compared the outcomes of US-guided RFA, apart from its feasibility, by matching the two treatment groups. Our results showed that the two groups had similar LTP, IDR, and RFS outcomes after treatment. Therefore, if US-guided RFA is not feasible, TACE + RFA under fluoroscopic guidance is an effective alternative, especially if surgical resection is infeasible. Thid may expand the treatment boundaries of ablation if conventional ablation and resection are not feasible for patients with very early or early HCC. 

Many previous studies have suggested better outcomes when TACE + RFA is performed compared with when RFA is performed alone [9,10,11,20,21,22]. While most of these studies evaluated treatment outcomes of TACE + RFA compared with those of RFA alone for intermediate-sized HCCs, our study included small tumors (≤3 cm) for comparison. Meanwhile, a previous study comparing TACE + RFA and RFA for early (≤3 cm) HCCs reported that TACE + RFA showed better outcomes than those of RFA alone [21]. Notably, the LTP rates after RFA were 14.3%, 32.3%, and 36.5% at 1, 3, and 5 years, respectively. Although IDR rates were not specifically mentioned, they appeared to be high, being >40% at 1 year and >60% at 2 years. In contrast, in our study, LTP rates were 7.4% at 1 and 2 years, and 10.5% at 5 years for RFA, and 0.0% at 1 year, 10.5% at 2 years, and 16.9% at 5 years for TACE + RFA. Further, IDR rates were 10.0% at 1 year, 25.9% at 2 years, and 53.0% at 5 years. Given that the study periods varied significantly between the previous study and ours (2007–2014 vs. 2012–2016), the lower LTP and IDR rates after RFA alone in our study may be attributed to recent advances in RFA techniques, including centripetal RFA using multiple electrodes, which enable larger and more efficient ablation. In addition, RFA outcomes have continuously improved with image fusion as it allows for more accurate lesion localization and targeting. This, in turn, has minimized the need for a second ablation session, especially when treating tumors in challenging locations. 

US-guided RFA was infeasible for tumors in the TACE + RFA group, implying that these tumors were more likely to be deeply positioned in challenging locations. Therefore, meticulous US monitoring during ablation would be challenging, even though the electrode could be inserted primarily under fluoroscopic guidance. Such suboptimal procedure monitoring may negatively affect treatment outcomes. In addition, fusion imaging was not performed during TACE + RFA. Therefore, a direct comparison between TACE + RFA and US-guided RFA may be difficult, even with PS matching. Nevertheless, our findings suggest that TACE + RFA can play a pivotal role in managing HCCs ≤ 3 cm, as treatment outcomes achieved through this procedure were comparable to those of general RFA.

There have been many advances in the field of locoregional treatments for HCC. For ablation, RFA has advanced significantly beyond its conventional methods. The cutting-edge method entails the utilization of multiple electrodes [23], embracing diverse energy modes such as dual-switching monopolar energy [24] and switching bipolar energy [25], and integrating new ablation strategies such as no-touch centripetal ablation [26]. Notably, the outcomes of US-guided RFA have improved significantly with the use of fusion imaging, CEUS, artificial ascites, and artificial pleural effusion, optimizing precision and efficacy. Recently, microwave ablation has surfaced as a potent modality for hepatic tumor ablation, especially for large tumors [27]. The comparable outcomes of TACE + RFA compared with RFA alone have the potential to expand the scope of ablation as a treatment option for HCC. Simultaneously, innovative therapeutic protocols to enhance outcomes in intermediate-stage HCCs have arisen, including the use of drug-eluting beads, TACE [28,29], and Y90 transarterial radioembolization [30]. While further investigations are warranted to ascertain the potential superiority of these new strategies over conventional TACE, they hold promise for inducing broader tumor necrosis and extending progression-free survival.

Recently, a prior study demonstrated a predictive relationship between post-treatment transient transaminase elevation and objective response to superselective conventional TACE [31]. Given that transaminases are generated by both hepatocytes and hepatocyte-derived tumor cells, observed transient transaminase elevation coupled with the absence of deterioration in a liver functional reserve could potentially imply that the elevation in serum enzyme concentrations emanates from tumors themselves, thus explaining the correlation with tumor response. Although this could not be evaluated in our study since transaminase was not routinely performed after the procedure, this association between transient hypertransaminasemia and treatment response has the potential to offer prognostic value after TACE + RFA as well, and also possibly after RFA. Further exploration into whether or not transient hypertransaminasemia could serve as a predictive factor for treatment response is warranted.

Meanwhile, a previous meta-analysis suggested that bland embolization with lipiodol showed no significant difference compared with conventional chemoembolization [32]. Likewise, bland embolization with lipiodol may be as effective as chemoembolization for combined treatment with RFA. However, in our institution, TACE is a standardized procedure involving the use of doxorubicin mixed with lipiodol, and thus this could not be assessed.

Our study had several limitations. Firstly, it was a retrospective study conducted at a single medical center, and direct validation data of TACE + RFA are not available. Additionally, the low LTP rate observed in our study might not be representative of all cases. Second, the study was carried out in a population predominantly affected by hepatitis B virus (HBV) infection. As a result, the findings may not be applicable to populations where HBV infection is not the prevailing cause of liver disease. Third, this study included a small number of patients. This was inevitable because TACE + RFA is not the first-line therapy for small HCCs at our medical center. To mitigate this issue, we utilized PSM to enhance group comparability and reduce potential bias. Moreover, despite the inherent imbalance in the small group sizes, we carried out a comprehensive multivariable analysis to gain a deeper understanding of the distinctions between the two groups and support the credibility of our findings. The results obtained from this multivariate analysis provided us with a higher degree of confidence in our primary conclusions. Therefore, this limitation may not compromise our main conclusions. Fourth, the feasibility of ablation under unenhanced CT guidance was not evaluated. However, at our medical center, ablation under CT guidance is usually not performed, and combined therapy with TACE under fluoroscopy is preferred if ablation is infeasible under US guidance. Although ablation under CT guidance offers the benefit of real-time volumetric assessment, potentially reducing the likelihood of requiring a subsequent treatment session in cases of incomplete ablation, fluoroscopy has several advantages over CT, such as less radiation exposure, shorter procedure times for needle placement, and the possibility to use a steep oblique approach to liver dome lesions to spare the thorax [33]. In addition, HCCs that are invisible upon unenhanced CT are common; therefore, this may not compromise the main results of our study, which showed the effectiveness of TACE + RFA for HCC that is infeasible for US-guided RFA.

## 5. Conclusions

In conclusion, the outcomes of TACE + RFA for HCC were similar to the general outcomes of US-guided RFA. Fluoroscopy-guided TACE + RFA may be an effective treatment when conventional US-guided RFA is not feasible.

## Figures and Tables

**Figure 1 cancers-15-05193-f001:**
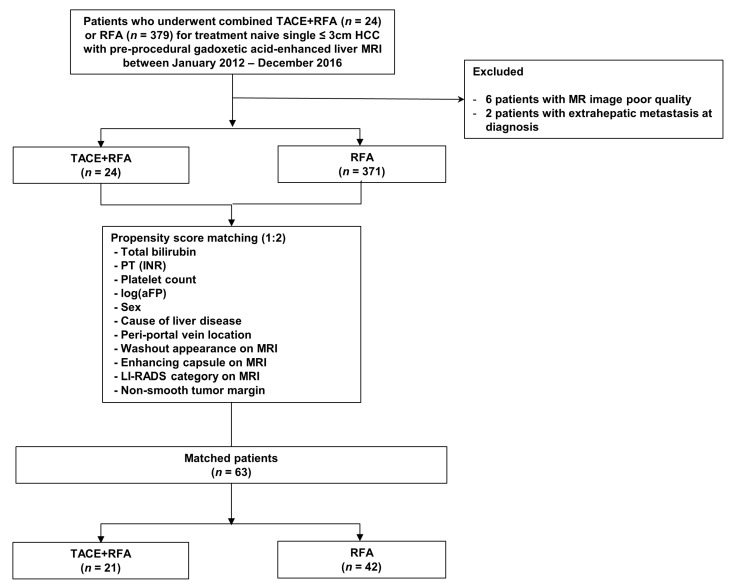
Patient inclusion process.

**Figure 2 cancers-15-05193-f002:**
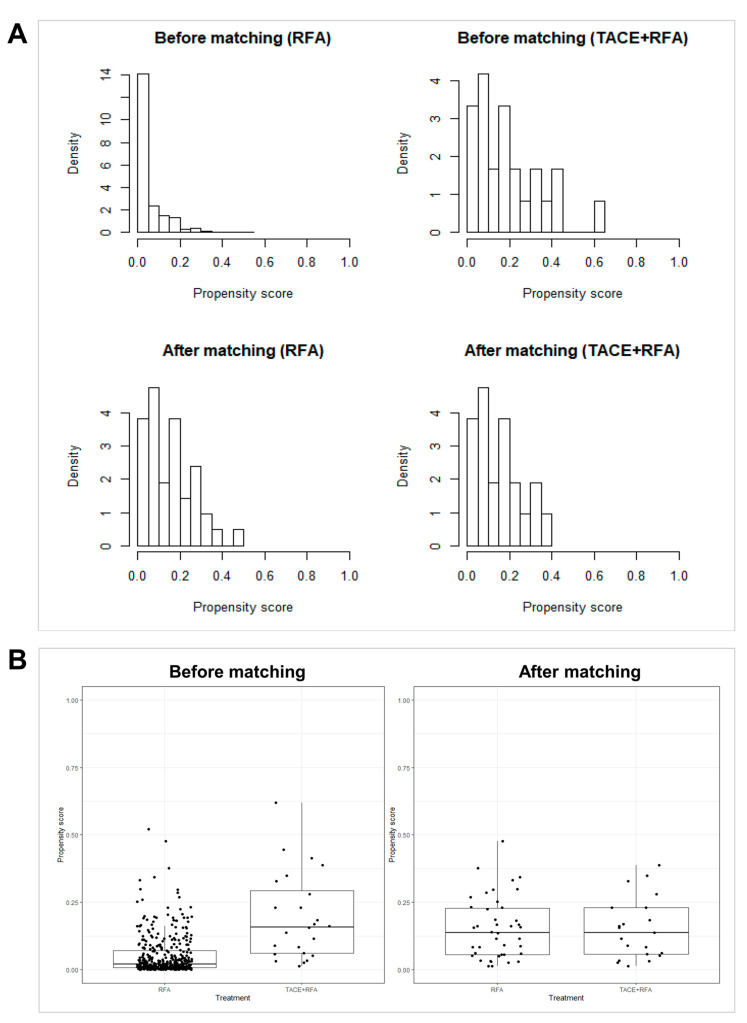
Histogram (**A**) and jitter plot (**B**) illustrating the distribution of propensity scores before and after propensity score matching between the TACE + RFA group and the RFA group.

**Figure 3 cancers-15-05193-f003:**
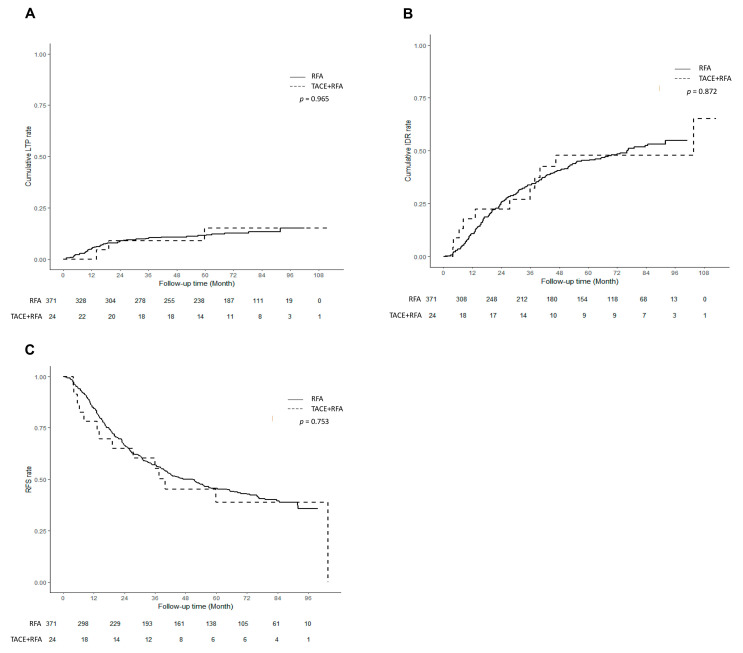
Outcomes after combined transarterial chemoembolization (TACE) and radiofrequency ablation (RFA) for hepatocellular carcinoma (HCC) in comparison with those of standard RFA for HCC before propensity score matching. (**A**) Cumulative incidence curves of local tumor progression after TACE + RFA compared with those of standard RFA for HCC. (**B**) Cumulative incidence curves of intrahepatic distant recurrence after TACE + RFA compared with those of standard RFA for HCC. (**C**) Recurrence-free survival curve after TACE + RFA compared with that of standard RFA for HCC.

**Figure 4 cancers-15-05193-f004:**
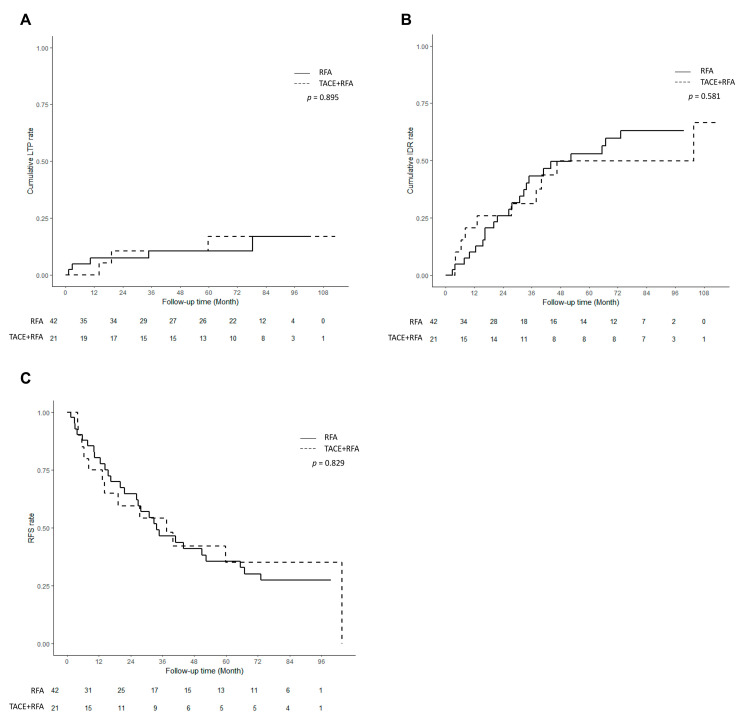
Outcomes after combined transarterial chemoembolization (TACE) and radiofrequency ablation (RFA) for hepatocellular carcinoma (HCC) in comparison with those of standard RFA for HCC after propensity score matching. (**A**) Cumulative incidence curves of local tumor progression after TACE + RFA compared with those of standard RFA for HCC. (**B**) Cumulative incidence curves of intrahepatic distant recurrence after TACE + RFA compared with those of standard RFA for HCC. (**C**) Recurrence-free survival curve after TACE + RFA compared with that of standard RFA for HCC.

**Figure 5 cancers-15-05193-f005:**
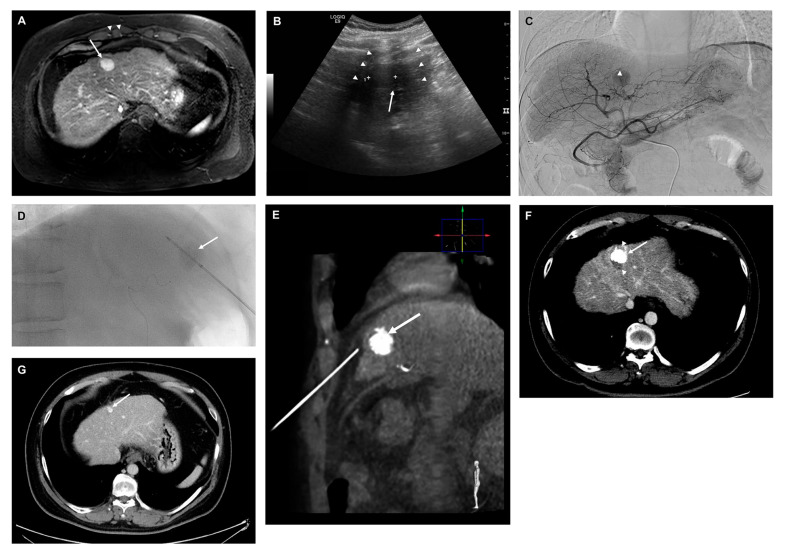
A 40-year-old male with chronic hepatitis B virus-associated chronic liver disease. (**A**) A 2 cm sized hepatocellular carcinoma (arrow) is seen in liver segment 4. Notice the location of the tumor, which is located posteriorly to the right costochondral junction (arrowhead). (**B**) Observation of a subtle hypoechoic lesion which had a very poor sonic window due to posterior acoustic shadowing (arrowheads) from the overlying costochondral junction upon planning ultrasound to evaluate the feasibility of ultrasound (US)-guided radiofrequency ablation (RFA. The lesion was regarded as infeasible for conventional US-guided RFA. (**C**) Patient underwent fluoroscopy guided-transarterial chemoembolization (TACE) combined with RFA (TACE + RFA). On the hepatic angiogram, nodular tumor staining is noted in liver segment 4 (arrow). Embolization using a mixture of 10 mg of adriamycin and 2 mL of iodized oil was performed after superselecting the feeding vessel. (**D**,**E**) Insertion of a 17-G RFA needle (2 cm active tip) into the tumor (arrow) immediately afterward under conventional fluoroscopic and cone-beam computed tomography (CT) guidance. (**F**) Liver CT obtained one day after TACE + RFA showing compact lipiodol uptake within the target lesion (arrow) and peritumoral ablation zone (arrowhead). It was regarded as a technical success. (**G**) Liver CT obtained 85 months after TACE + RFA showing that a small amount of iodized oil remained within the shrunken treated tumor (arrow). There was no evidence of local tumor progression or other kinds of recurrence.

**Table 1 cancers-15-05193-t001:** Patient characteristics.

	TACE + RFA (n = 24)	RFA (n = 371)	*p*-Value
Age (year) *	57.5 (39–77)	58 (31–85)	0.588
Sex (male)	22 (91.7)	272 (73.3)	0.079
Cause of liver disease			0.023
HBV	12 (50)	279 (75.2)	
HCV	5 (20.8)	38 (10.2)	
Alcohol	6 (25)	35 (9.4)	
Others	1 (4.2)	19 (5.1)	
Albumin (mg/dL)	4.2 (1.9–5.1)	4 (2.8–4.6)	0.159
Bilirubin (mg/dL)	0.6 (0.2–4.5)	0.7 (0.3–2.7)	0.123
Prothrombin time (INR)	1.1 (0.9–1.6)	1.2 (1–1.5)	0.092
AST (U/L)	33 (10–552)	33 (18–88)	0.299
Platelet count (K)	109 (50–475)	87.5 (27–290)	0.102
ALBI grade			0.433
1	14 (58.3)	253 (68.2)	
2	10 (41.7)	112 (30.2)	
3	0 (0)	6 (1.6)	
APRI *	0.891 (0.422–8.148)	0.786 (0.145–12.545)	0.425
Child–Pugh classification			0.907
A	23 (95.8)	345 (93)	
B	1 (4.2)	26 (7)	
log(AFP) (ng/mL) *	1.75 (0.74–3.28)	2.10 (0.26–7.70)	<0.001
PIVKA-II (mAU/mL]) *	27 (12–268)	22 (9–11,078)	0.549
MoRAL score *	62.7 (41.6–190.4)	61.6 (36.3–1164.6)	0.974
Tumor size (cm) *	1.5 (1–2.7)	1.6 (1–2.9)	0.743
Tumor location			
Periportal vein	5 (20.8)	21 (5.7)	0.013
Perihepatic vein	1 (4.2)	33 (8.9)	0.671
Subcapsular (reference = non-subcapsular)	11 (45.8)	129 (34.8)	0.510
Arterial phase hyperenhancement			0.319
No	2 (8.3)	13 (3.5)	
Non-rim	21 (87.5)	319 (86)	
Rim	1 (4.2)	39 (10.5)	
Washout appearance	8 (33.3)	194 (52.3)	0.112
Enhancing capsule	5 (20.8)	155 (41.8)	0.070
Peripheral washout	0 (0)	4 (1.1)	1.000
Progressive enhancement	0 (0)	13 (3.5)	0.732
Transitional phase targetoid	0 (0)	9 (2.4)	0.947
Hepatobiliary phase targetoid	0 (0)	12 (3.2)	0.779
Diffusion weighted image targetoid	1 (4.2)	29 (7.8)	0.792
Non-targetoid LR-M feature	0 (0)	0 (0)	-
LI-RADS category			0.006
3	13 (54.2)	88 (23.7)	
4	4 (16.7)	56 (15.1)	
5	6 (25)	166 (44.7)	
M	1 (4.2)	61 (16.4)	
Peritumoral enhancement	2 (8.3)	74 (19.9)	0.258
Non-smooth margin	2 (8.3)	103 (27.8)	0.064
Peritumoral hypointensity	1 (4.2)	31 (8.4)	0.732
Low SI on hepatobiliary phase (reference = iso/high)	21 (87.5)	354 (95.4)	0.217
MVI high-risk group	1 (4.2)	31 (8.4)	0.655

Note: Unless stated otherwise, the data are presented as the number of patients (lesions) with percentages in parentheses. * Data are expressed as medians with ranges in parentheses. HBV, hepatitis B virus; HCV, hepatitis C virus; INR, international normalized ratio; AST, aspartate aminotransferase; ALBI grade, albumin–bilirubin grade; APRI, AST/platelet ratio index; AFP, alpha-fetoprotein; PIVKA-II, protein induced by vitamin K absence-II; MoRAL score, Model for Tumor Recurrence After Living Donor Liver Transplantation score; LR-M, LI-RADS category M; LI-RADS, Liver Imaging Reporting and Data System; SI, signal intensity; MVI, microvascular invasion; TACE, transarterial chemoembolization; RFA, radiofrequency ablation.

**Table 2 cancers-15-05193-t002:** Baseline characteristics of study patients after propensity score matching, and balance check before and after propensity score matching for matched variables.

	After Propensity Score Matching			Before Matching TACE + RFA (*n* = 24), RFA (*n* = 371)		After Matching TACE + RFA (*n* = 21), RFA (*n* = 42)	
Variables	TACE + RFA (n = 21)	RFA (n = 42)	*p*	SMD	*p*	SMD	*p*
Total bilirubin, mg/dL *	0.70 (0.3–2.7)	0.70 (0.2–4.5)	0.546	0.357	0.123	0.166	0.591
Prothrombin time, INR *	1.12 (0.99–1.51)	1.17 (0.96–1.6)	0.741	0.376	0.092	−0.090	0.768
Platelet count, ×10^9^/L *	89.00 (27–290)	88.00 (52–241)	0.938	−0.357	0.102	0.017	0.948
log(AFP) *	1.82 (0.74–3.28)	1.71 (0.26–5.73)	0.723	−0.498	0.001	−0.131	0.727
Sex	38 (66.7%)	19 (90.5%)	1.000	−0.494	0.006	0.000	1.000
Cause of liver disease							
HBV	12 (57.1%)	25 (44.6%)	0.953	−0.533	0.026	−0.047	0.865
HCV	5 (23.8%)	9 (16.1%)		0.291	0.230	0.055	0.857
Alcohol	3 (14.3%)	7 (12.5%)		0.415	0.102	−0.066	0.792
Others	1 (4.8%)	1 (1.8%)		−0.045	0.827	0.109	0.686
Periportal vein location	3 (14.3%)	7 (12.5%)	0.790	0.452	0.089	−0.066	0.816
Washout appearance (PVP)	7 (33.3%)	11 (19.6%)	0.544	−0.386	0.073	0.148	0.610
Enhancing capsule	4 (19.0%)	11 (19.6%)	0.510	−0.459	0.025	−0.178	0.560
LI-RADS category							
3	11 (52.4%)	23 (41.1%)	0.919	0.649	0.008	−0.047	0.873
4	4 (19.0%)	10 (17.9%)		0.043	0.846	−0.118	0.686
5	5 (23.8%)	8 (14.3%)		−0.419	0.045	0.109	0.686
M	1 (4.8%)	1 (1.8%)		−0.410	0.011	0.109	0.686
Non-smooth tumor margin	2 (9.5%)	3 (5.4%)	0.743	−0.519	0.004	0.079	0.792

Note: Unless otherwise indicated, data are the number of patients (lesions) with percentages in parentheses. * Data are medians with ranges in parentheses. INR, international normalized ratio; AFP, alpha-fetoprotein; HBV, hepatitis B virus; HCV, hepatitis C virus; PVP, portal venous phase; LR, LI-RADS, Liver Imaging Reporting and Data System category; SMD, standard mean difference; TACE, transarterial chemoembolization; RFA, radiofrequency ablation.

**Table 3 cancers-15-05193-t003:** Univariable and multivariable analysis for outcomes for all patients.

	Local Tumor Progression				Intrahepatic Distant Recurrence				Recurrence-Free Survival			
	Univariable Analysis		Multivariable Analysis		Univariable Analysis		Multivariable Analysis		Univariable Analysis		Multivariable Analysis	
	HR (95% CI)	*p*	HR (95% CI)	*p*	HR (95% CI)	*p*	HR (95% CI)	*p*	HR (95% CI)	*p*	HR (95% CI)	*p*
TACE + RFA [RFA]	1.026 (0.319–1.026)	0.965			0.949 (0.501–0.949)	0.872			1.094 (0.625–1.094)	0.753		
Age	1.016 (0.987–1.016)	0.277			1.012 (0.996–1.012)	0.143	1.023 (1.006–1.023)	0.009	1.022 (1.008–1.022)	0.002	1.025 (1.009–1.025)	0.002
Male [Female]	0.865 (0.430–0.865)	0.685			0.86 (0.6–0.86)	0.409			1.047 (0.771–1.047)	0.767		
Liver disease [HBV]												
HCV	1.384 (0.581–1.384)	0.463			1.513 (0.943–1.513)	0.086			2.303 (1.575–2.303)	<0.001	1.838 (1.182–1.838)	0.007
Alcohol	0.920 (0.326–0.920)	0.875			1.088 (0.663–1.088)	0.737			1.253 (0.806–1.253)	0.317	0.921 (0.553–0.921)	0.753
Others	1.366 (0.419–1.366)	0.605			1.718 (0.95–1.718)	0.074			1.551 (0.879–1.551)	0.130	1.181 (0.628–1.181)	0.605
Albumin (mg/dL)	0.946 (0.541–0.946)	0.845			0.486 (0.373–0.486)	<0.001	0.477 (0.338–0.477)	<0.001	0.474 (0.374–0.474)	<0.001	0.501 (0.370–0.501)	<0.001
Bilirubin (mg/dL)	0.871 (0.486–0.871)	0.641			1.56 (1.203–1.56)	0.001			1.532 (1.207–1.532)	<0.001	1.350 (0.992–1.35)	0.056
ALBI grade												
Grade 2	1.232 (0.663–1.232)	0.509			2.019 (1.486–2.019)	<0.001			2.189 (1.661–2.189)	<0.001		
Grade 3	4.255 (0.797–4.255)	0.090			7.271 (2.269–7.271)	0.001			5.070 (1.595–5.07)	0.006		
PT (INR)	0.642 (0.083–0.642)	0.671			4.633 (1.894–4.633)	0.001			5.047 (2.199–5.047)	<0.001		
AST (U/L)	0.998 (0.987–0.998)	0.650			1.004 (1.002–1.004)	<0.001			1.004 (1.002–1.004)	<0.001	1.009 (1.000–1.009)	0.045
Platelet count (K)	0.999 (0.993–0.999)	0.603			0.995 (0.992–0.995)	0.005	0.997 (0.993–0.997)	0.096	0.996 (0.993–0.996)	0.006	0.995 (0.991–0.995)	0.030
APRI	0.980 (0.757–0.98)	0.881			1.194 (1.101–1.194)	<0.001			1.169 (1.084–1.169)	<0.001	0.710 (0.492–0.71)	0.069
log(AFP) (ng/mL)	1.039 (0.878–1.039)	0.653			1.116 (1.027–1.116)	0.010			1.104 (1.023–1.104)	0.011		
Child–Pugh B [reference = A]	1.086 (0.337–1.086)	0.890			2.494 (1.53–2.494)	<0.001	1.695 (0.956–1.695)	0.071	2.291 (1.442–2.291)	<0.001		
PIVKA-II	1.000 (0.999–1.000)	0.779			1.000 (1.000–1.000)	0.077			1.000 (1.000–1.000)	0.141		
MoRAL > 68	1.190 (0.658–1.190)	0.565			1.559 (1.144–1.559)	0.005	1.484 (1.051–1.484)	0.025	1.494 (1.128–1.494)	0.005	1.475 (1.099–1.475)	0.010
Tumor size	1.488 (0.78–1.488)	0.227			1.533 (1.091–1.533)	0.014	1.448 (0.995–1.448)	0.053	1.552 (1.145–1.552)	0.005	1.653 (1.190–1.653)	0.003
Peri-PV	1.415 (0.508–1.415)	0.507			0.967 (0.525–0.967)	0.913			0.937 (0.535–0.937)	0.821		
Peri-HV	1.021 (0.366–1.021)	0.969			0.682 (0.371–0.682)	0.220			0.918 (0.566–0.918)	0.727		
Subcapsular	1.929 (1.088–1.929)	0.024	1.898 (1.071–1.898)	0.028	1.041 (0.767–1.041)	0.795			1.140 (0.866–1.14)	0.351		
APHE [No]												
Non-rim	1.654 (0.227–1.654)	0.620			1.533 (0.629–1.533)	0.347			1.395 (0.656–1.395)	0.387		
Rim	3.128 (0.391–3.128)	0.283			0.936 (0.334–0.936)	0.900			1.087 (0.456–1.087)	0.851		
Washout app.	1.386 (0.774–1.386)	0.273			1.051 (0.782–1.051)	0.740			0.967 (0.741–0.967)	0.808		
Enhancing capsule	1.216 (0.684–1.216)	0.506			1.310 (0.975–1.31)	0.073	1.536 (1.115–1.536)	0.009	1.103 (0.842–1.103)	0.475		
Peripheral washout	1.995 (0.274–1.995)	0.495			1.813 (0.579–1.813)	0.307			1.360 (0.435–1.36)	0.597		
Progressive Enhancement	1.958 (0.608–1.958)	0.26			0.592 (0.220–0.592)	0.300			0.747 (0.332–0.747)	0.481		
TP targetoid	0.812 (0.112–0.812)	0.837			0.589 (0.188–0.589)	0.364			0.446 (0.143–0.446)	0.166		
HBP targetoid	2.260 (0.701–2.260)	0.172			0.651 (0.241–0.651)	0.396			0.640 (0.264–0.640)	0.325		
DWI targetoid	1.799 (0.764–1.799)	0.179			1.048 (0.607–1.048)	0.865			1.061 (0.646–1.061)	0.815		
Non-targetoid LR-M	1.907 (0.989–1.907)	0.054	1.859 (0.964–1.859)	0.064	0.785 (0.51–0.785)	0.270			0.864 (0.590–0.864)	0.451		
LR category												
4	1.106 (0.393–1.106)	0.849			1.150 (0.711–1.15)	0.569			1.050 (0.685–1.050)	0.822		
5	1.271 (0.578–1.271)	0.551			1.194 (0.822–1.194)	0.353			1.027 (0.735–1.027)	0.876		
M	2.212 (0.932–2.212)	0.072			0.885 (0.534–0.885)	0.634			0.884 (0.567–0.884)	0.586		
Peritumoral enhancement	1.847 (0.988–1.847)	0.054			1.426 (1.011–1.426)	0.043			1.215 (0.879–1.215)	0.238		
Non-smooth margin	1.465 (0.801–1.465)	0.215			1.058 (0.761–1.058)	0.736			1.026 (0.760–1.026)	0.865		
Peritumoral hypointensity	1.047 (0.376–1.047)	0.931			1.189 (0.721–1.189)	0.497			0.896 (0.546–0.896)	0.665		
Low SI on HBP [iso/high]	0.937 (0.227–0.937)	0.928			0.853 (0.400–0.853)	0.679			0.967 (0.512–0.967)	0.917		
MVI-high risk	0.998 (0.357–0.998)	0.997			1.151 (0.697–1.151)	0.583			0.967 (0.596–0.967)	0.891		

TACE, transarterial chemoembolization; RFA, radiofrequency ablation; HBV, hepatitis B virus; HCV, hepatitis C virus; ALBI grade, albumin–bilirubin grade; PT, prothrombin time; INR, international normalized ratio; AST, aspartate aminotransferase; APRI, AST/platelet ratio index; AFP, alpha-fetoprotein; MoRAL score, Model for Tumor Recurrence After Living Donor Liver Transplantation score; PV, portal vein; HV, hepatic vein; APHE, arterial phase hyperenhancement; app., appearance; TP, transitional phase; HBP, hepatobiliary phase; DWI, diffusion weighted images; LR-M, LI-RADS category M; LI-RADS, Liver Imaging Reporting and Data System; SI, signal intensity; MVI, microvascular invasion; HR, hazard ratio; CI, confidence interval.

## Data Availability

The data that support the findings of this study are not publicly available as they contain information that could compromise the privacy of research participants but may be made available from the corresponding authors (M.W.L.; D.H.) upon reasonable request.

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
