# Peer review of "Combined Transarterial Chemoembolization and Radiofrequency Ablation for Hepatocellular Carcinoma Infeasible for Ultrasound-Guided Percutaneous Radiofrequency Ablation: A Comparative Study with General Ultrasound-Guided Radiofrequency Ablation Outcomes"

_cancers, 2023, doi:10.3390/cancers15215193_

Round 1

Reviewer 1 Report (Previous Reviewer 2)

Comments and Suggestions for Authors

Very interesting study. Major limitation is the unbalanced sample size between the two cohorts but the propensity score matching obviate at least partially to this issue.

Figures and tables are fine.

The authors should update a little their bibliography. For example, replace ref 31 with a more recent MA by the same authors (PMID: 33339274 ).

I recommend to add a comment on the impact of these multimodality treatments on post-recurrence survival after RFA (cite the series PMID: 25085684)

Author Response

Reviewer 1

Comments and Suggestions for Authors

Very interesting study. Major limitation is the unbalanced sample size between the two cohorts but the propensity score matching obviate at least partially to this issue.

Figures and tables are fine.

R1-1. The authors should update a little their bibliography. For example, replace ref 31 with a more recent MA by the same authors (PMID: 33339274 ).

  1. A) Thank you for the recommendation. The reference has been changed to a newer one.

R1-2. I recommend to add a comment on the impact of these multimodality treatments on post-recurrence survival after RFA (cite the series PMID: 25085684)

  1. A) We appreciate your suggestion to cite the series PMID: 25085684; however, we would like to provide some context regarding our reluctance to include this reference. The referenced paper primarily focuses on the impact of recurrence type (LTP, IDR, EM, or PVTT) and Child-Pugh liver function on post-recurrence survival.

While we understand the importance of considering post-recurrence survival, we believe that the scope and context of our study are somewhat distinct, making it challenging to directly relate our findings to those presented in PMID: 25085684. Including this reference in our manuscript may, in some instances, appear forced or tangential to our core message.

We hope you can understand our perspective on this matter. If you still strongly recommend its inclusion, we are open to further discussion on how to incorporate it in a way that enhances the overall quality and relevance of our paper.

Reviewer 2 Report (Previous Reviewer 3)

Comments and Suggestions for Authors

This article is improving for reviewers' requests.

Comments on the Quality of English Language

I have no comments.

Author Response

Reviewer 2

I have no comments.

  1. A) Thank you for your kind review. It is greatly appreciated.

Reviewer 3 Report (New Reviewer)

Comments and Suggestions for Authors

- Please include in the abstract the possibility and data that is available to treat patients with MRI fusion in ultrasound and CT. It also need to be discussed in the discussion that this allows a volumetric assessment after the ablation that is not available after ultrasound guided ablation. in addition this allow to treat the patient with inly 1 intervention.

- The fact that previous embolisation enhances the abaltion zone is correct, but has nothing to do that invisible tumor can be treated by fluorescence guidance. Please do not mix up these 2 aspects. In particular not for the treatment of small HCCs < 3 cm, where the combination is usually not performed.

2.4 why was RFA monotherapy performed with MRI or CT fusion and additional CEUS in some cases if in this group the tuors should have been visible on ultrasound?

Was tumor size and location including proximity to important structures not considered in the matching?

Figure 5: please show the pre-intervention CT, was the tumor invisible there?

Please discuss the availabilty of navigation systems allowing to treat tumors in challenging locations with a single intervention.

Please discuss that no direct validation of the treatment is avaialble

Please discuss the value of your statistical analyses in such a small group of patients, in particular multivariate analyses.

Please shorten the discussion

Conclusion is too strong since other techniques are available and the data is to weak to suggest this

Comments on the Quality of English Language

No particular comments

Author Response

Reviewer 4 Report (New Reviewer)

Comments and Suggestions for Authors

Nice review of the subject.

Can you please explain why TACE was chosen instead of bland embolization with lipoidal.

Can you please explain what type of TACE was performed. Was the drug and procedure standardized.

Can you please explain why conventional TACE was chosen over Deb TACE.

Release review some of the these references:

Thornton LM, Cabrera R, Kapp M, Lazarowicz M, Vogel JD, Toskich BB. Radiofrequency vs Microwave Ablation After Neoadjuvant Transarterial Bland and Drug-Eluting Microsphere Chembolization for the Treatment of Hepatocellular Carcinoma. Curr Probl Diagn Radiol. 2017 Nov-Dec;46(6):402-409. doi: 10.1067/j.cpradiol.2017.02.006. Epub 2017 Feb 20. PMID: 28392205; PMCID: PMC5563480.

Facciorusso A, Bellanti F, Villani R, Salvatore V, Muscatiello N, Piscaglia F, Vendemiale G, Serviddio G. Transarterial chemoembolization vs bland embolization in hepatocellular carcinoma: A meta-analysis of randomized trials. United European Gastroenterol J. 2017 Jun;5(4):511-518. doi: 10.1177/2050640616673516. Epub 2016 Oct 3. PMID: 28588882; PMCID: PMC5446148.

Author Response

Reviewer 4.

Comments and Suggestions for Authors

Nice review of the subject.

R4-1. Can you please explain why TACE was chosen instead of bland embolization with lipoidal.

  1. A) We appreciate your question. The selection of TACE over bland embolization with lipiodol in our study stems from several considerations. It's noteworthy that a meta-analysis has demonstrated the non-superiority of TACE compared to bland embolization in HCC patients (PMID: 28588882, suggested reference from R4-4). While bland embolization with lipiodol could have been a viable option for subsequent RFA, it's essential to note that TACE is a standardized procedure using doxorubicin mixed with lipiodol in our institution. Our institution often reserves TACE as a secondary treatment option when primary treatment modalities such as ablation or resection are not feasible which aligns with the updated BCLC guideline recommendations. Therefore, as embolization is a standardized procedure using a mixture of doxorubicin and lipiodol, combined treatment with blind embolization could not be assessed. I hope you understand our situation. The following has been added to the revised manuscript.

Meanwhile, a previous meta-analysis suggested that bland embolization with lipiodol showed no significant difference with conventional chemoembolization (36). Likewise, bland embolization with lipiodol may be as effective as chemoembolization for combined treatment with RFA. However, in our institution, TACE is a standardized procedure using doxorubicin mixed with lipiodol, and thus this could not be assessed.

R4-2. Can you please explain what type of TACE was performed. Was the drug and procedure standardized.

  1. A) I am sorry the procedure details for TACE are missing. We have added the following in the revised manuscript. Further details have been added to the supplementary materials.

Selective TACE was carried out via a microcatheter by transarterially infusing a mixture of 2–5 mL of iodized oil (Lipiodol; Laboratoire Andre Guerbet, Aulnay-sous-Bois, France) and 10–20 mg of doxorubicin hydrochloride (Adriamycin; Dong-A Pharm, Seoul, Korea). The iodized oil and doxorubicin was emulsified by vigorous pumping (10–20 times) between two syringes connected with a three-way stopcock just before infusion. The specific amount of iodized oil and doxorubicin administered depended on tumor size and vascularity. Following the transarterial infusion of this mixture, embolization of the tumor's feeding artery was executed using gelatin sponge pledgets (Cutanplast; Mascia Brunelli, Milan, Italy), which were manually cut into ~1 mm3 pieces. Embolization continued until blood flow within the tumor’s feeding artery ceased.

R4-3.Can you please explain why conventional TACE was chosen over Deb TACE.

  1. A) The treatment outcomes of combined RFA and DEB-TACE is very interesting. However, the main purpose of our study was to evaluate the outcomes of TACE+RFA for HCCs that were invisible on US. Lipiodol from TACE will be visible on fluoroscopy which will enable ablation by targeting the lipiodol. If DEB-TACE was performed, ablation would not have been possible since it the lesion would not be visible on fluoroscopy after DEB-TACE.

R4-4. Release review some of the these references:

Thornton LM, Cabrera R, Kapp M, Lazarowicz M, Vogel JD, Toskich BB. Radiofrequency vs Microwave Ablation After Neoadjuvant Transarterial Bland and Drug-Eluting Microsphere Chembolization for the Treatment of Hepatocellular Carcinoma. Curr Probl Diagn Radiol. 2017 Nov-Dec;46(6):402-409. doi: 10.1067/j.cpradiol.2017.02.006. Epub 2017 Feb 20. PMID: 28392205; PMCID: PMC5563480.

Facciorusso A, Bellanti F, Villani R, Salvatore V, Muscatiello N, Piscaglia F, Vendemiale G, Serviddio G. Transarterial chemoembolization vs bland embolization in hepatocellular carcinoma: A meta-analysis of randomized trials. United European Gastroenterol J. 2017 Jun;5(4):511-518. doi: 10.1177/2050640616673516. Epub 2016 Oct 3. PMID: 28588882; PMCID: PMC5446148.

  1. A) As discussed in R4-1, DEB-TACE may be beyond the scope of our study. Instead, we have added the second reference (bland embolization vs. chemoembolization) to the revised manuscript.

Round 2

Reviewer 1 Report (Previous Reviewer 2)

Comments and Suggestions for Authors

The revised version of the paper is OK. Thank you!

This manuscript is a resubmission of an earlier submission. The following is a list of the peer review reports and author responses from that submission.

Round 1

Reviewer 1 Report

Comments and Suggestions for Authors

 The authors have suggested that fluoroscopy- guided TACE combined with RFA may be the alternative therapy for small HCCs. This therapy is attractive and  the the therapeutic outcomes are comparable to those of conventional RFA therapy. But I have some concerns with this therapy.

1. How long does it take for one treatment? It would take longer than the laparoscopic surgery. There may be concern that the physical burden on the patient may be severe.

2. Is radiation exposure excessive both patient and medical staffs? 

Reviewer 2 Report

Comments and Suggestions for Authors

Very interesting and well conducted study. After propensity score matching many patients were missed but the analysis resulted much more balanced.

My comments:

1) Which variables were used for PS matching? The authors should report them in the statistical paragraph

2) The authors should report also the absolute number of the variables after matching

3) Among the variables tested in the regression analysis, it would be nice to see also the impact of post-TACE hypertransaminasemia on survival outcomes. If this data is not available, at least comment this aspect in the discussion citing the recent paper (PMID: 34683182 )

4) The authors should comment on the state of the art of loco-regional treatments (cite the recent MA: PMID: 27366304 )

5) Could the authors add the histogram and the jitter plot for PS matching?

Reviewer 3 Report

Comments and Suggestions for Authors

The author reported the TACE+RFA appears to be an effective treatment when US-guided RFA. Especially, in locations where RFA is difficult. The reviewers are basically of the same opinion. However, the author needs to address several issues before publication.

 Major

 (1).  The author should remove similar factors (Albumin, ALBI grade, Child-Pugh....) in multivariate analysis because they are confounding factors.

 (2).  How was the initial treatment effect of TACE + RFA?. How many patients achieve completed in one session in this study? If the RFA margin is not enough, was there an additional RFA after that?

Minor

(3).  Please indicate the percentage of the whole, not only males in sex in Table 1

(4).  How was the C statics in PSM?

Comments on the Quality of English Language

Minor editing of English language required